# Ferroptosis-Linked Six-Gene Panel Enables Machine Learning-Assisted Diagnosis and Therapeutic Guidance in Lung Adenocarcinoma

**DOI:** 10.3390/biology14091280

**Published:** 2025-09-17

**Authors:** Faris Alrumaihi

**Affiliations:** Department of Medical Laboratories, College of Applied Medical Sciences, Qassim University, Buraydah 51452, Saudi Arabia; f_alrumaihi@qu.edu.sa

**Keywords:** ferroptosis, lung adenocarcinoma, machine learning, gene signature, prognosis

## Abstract

Lung adenocarcinoma is the most common form of lung cancer, and many patients lack gene mutations that can be targeted by current therapies. To address this, we identified six genes with consistently altered activity in lung tumours compared to normal lung tissue. These genes help distinguish cancer from normal tissue, predict patient survival, and may guide new treatment strategies. Using machine learning, we showed that this small gene panel can accurately detect lung cancer. One gene, PLK1, showed a strong association with poor outcomes and could be a promising drug target. Although these genes were initially selected based on their link to a type of cell death called ferroptosis, our findings suggest they more broadly reflect cancer cell growth and treatment resistance. This work could help doctors better diagnose lung cancer and tailor treatments for patients who do not benefit from existing options.

## 1. Introduction

Lung adenocarcinoma (LUAD), the predominant histological subtype of non-small-cell lung cancer (NSCLC), accounts for nearly half of all lung cancer diagnoses worldwide and remains clinically challenging [1,2]. Despite the emergence of precision oncology and immunotherapy over the past decade, lung cancer remains the leading cause of cancer-related mortality, with five-year survival rates lingering below 20% in advanced-stage LUAD [3,4,5]. This persistent lethality stems not only from late-stage diagnosis in a substantial proportion of patients [6] but also from the eventual failure of targeted therapies and immune checkpoint inhibitors due to resistance mechanisms such as secondary mutations, pathway rewiring, and immune escape [7,8].

While genomic stratification based on EGFR, ALK, KRAS, BRAF, MET, and RET has improved treatment for some LUAD patients [9], these driver mutations are absent in many cases [10]. For these mutation-negative or treatment-resistant tumours, therapeutic options are limited, and predictive biomarkers remain elusive. Recent efforts have therefore begun to explore transcriptomic signatures as complementary tools for diagnosis and treatment planning. Unlike static DNA alterations, gene expression patterns provide a dynamic snapshot of tumour cell states, capturing features such as metabolic rewiring, oxidative stress, and immune evasion [11,12,13]. However, the clinical translation of expression-based classifiers has been hindered by limited reproducibility, high dimensionality, and a lack of functional anchoring to actionable pathways [14]. In this context, transcriptomic signatures are increasingly investigated as complementary classifiers, particularly for patients lacking actionable driver mutations.

Ferroptosis, a regulated form of iron-dependent cell death characterised by lipid peroxidation, has recently emerged as a potential therapeutic vulnerability in solid tumours [15]. By disrupting redox balance and membrane lipid homeostasis, ferroptosis inducers can synergise with chemotherapy, radiotherapy, and targeted agents to overcome resistance [16,17,18]. In LUAD, several studies have shown that ferroptosis modulation affects tumour progression and treatment response. For example, erastin and RSL3 were reported to resensitise resistant LUAD cells to therapy, and clinical data suggest ferroptosis regulators such as GPX4 and ACSL4 influence patient outcomes [19,20,21]. However, most investigations have focused narrowly on a few canonical regulators, without systematically evaluating ferroptosis-associated transcriptional signatures in patient cohorts. Notably, many ferroptosis-linked genes also regulate proliferation and the cell cycle, making it difficult to distinguish ferroptotic from mitotic processes in LUAD. This overlap raises the question of whether ferroptosis-seeded gene lists can yield expression panels that, even if not strictly ferroptotic in function, may still provide clinically useful biomarkers.

Beyond intrinsic tumour biology, ferroptosis also intersects with the tumour immune microenvironment. Lipid peroxidation products and damage-associated molecular patterns (DAMPs) released during ferroptotic death can stimulate dendritic cell maturation and antigen presentation, thereby enhancing CD8^+^ T-cell responses [22]. These findings highlight ferroptosis as a dual regulator of tumour progression and anti-tumour immunity, particularly relevant in LUAD, where immunotherapy has reshaped clinical practice. Several studies have begun to explore ferroptosis, reporting that ferroptosis-related gene expression signatures were associated with immune infiltration and survival outcomes [23]. Another study linked ferroptosis scores to targeted therapy sensitivity, including EGFR inhibitor response in LUAD [24]. Moreover, one study described a ferroptosis-related lncRNA panel that stratified prognosis and correlated with immune checkpoint gene expression [25]. While informative, these studies were either restricted to single data modalities or lacked therapeutic integration, underscoring the need for a compact, biologically grounded ferroptosis signature with direct clinical applicability.

To address this gap, we conducted an integrative transcriptomic and clinical analysis of LUAD using TCGA data, selecting candidate genes from a curated ferroptosis database. Through differential expression and survival filtering, we derived a compact six-gene panel. Functional enrichment and pathway scoring revealed that although seeded from ferroptosis-associated genes, the panel’s utility appears primarily linked to cell-cycle and proliferation pathways, with PLK1 emerging as the sole independent prognostic factor. The panel stratifies overall survival, demonstrates clear separation between tumour and adjacent normal tissue, and shows exploratory associations with drug sensitivity profiles. In particular, the prominence of PLK1 suggests a potential link to existing PLK1-targeted therapies. These findings support the development of compact, biologically anchored expression panels for LUAD, while also underscoring the need to disentangle ferroptosis from broader proliferative programs in future work.

## 2. Materials and Methods

### 2.1. Data Acquisition and Processing

RNA-sequencing data for LUAD were retrieved from The Cancer Genome Atlas (TCGA) via the TCGAbiolinks R package (v2.30.0) [26]. Gene-level raw counts generated using the STAR alignment workflow were downloaded, including both primary tumour and matched normal lung tissues [27]. In total, 598 samples were analysed, including 539 tumour and 59 normal tissue samples. Expression data were processed to generate a unified expression matrix, with consistent metadata annotations across all samples. The final dataset included raw gene expression counts and corresponding sample metadata, stored as both CSV files and a Summarized Experiment object for downstream analyses. All processing steps were performed using R (v4.2.0) [28], and data files were retained in the project directory to ensure reproducibility.

### 2.2. Differential Gene Expression Analysis

Differential gene expression analysis was performed using the DESeq2 package (v1.40.2) [29]. Raw count data and sample metadata were aligned to ensure consistent identifiers across datasets, and genes with fewer than 10 reads in ≥10% of samples were excluded. To assess both global and patient-specific expression changes, two complementary designs were implemented. For the unpaired analysis, all LUAD tumours (n = 539) were contrasted against all normal lung tissues (n = 59) using a design formula based solely on condition (Tumour vs. Normal). For the paired analysis, we restricted the cohort to patients with matched tumour–normal pairs (n = 58) and incorporated patient identity as a blocking factor in the design formula (~patient_id + condition), thereby controlling for inter-patient baseline variation. This allowed tumour expression profiles to be evaluated relative to their matched normal counterparts. In both models, dispersion was estimated using the DESeq2 framework, and log_2_ fold changes were moderated with apeglm shrinkage to improve interpretability. Ensembl gene identifiers were mapped to HGNC symbols via biomaRt (v2.56.1) [30], with duplicate mappings collapsed by averaging. Genes with adjusted *p* < 0.05 and absolute log_2_ fold change > 1 were considered significantly differentially expressed. Differentially expressed genes were further classified as upregulated or downregulated relative to normal tissues. Variance-stabilised transformed (VST) expression values were used to visualise the top 30 DEGs from each design in heatmaps annotated by sample condition, and volcano plots were generated to depict the genome-wide distribution of fold changes and significance.

### 2.3. Variance-Stabilising Transformation and Gene Symbol Annotation

Gene expression normalisation was performed using VST from the DESeq2 package. Raw count data and sample metadata were loaded, and samples were categorised as tumour or normal based on TCGA annotations. Sequencing depth differences were corrected by estimating size factors prior to transformation. VST was applied with condition-specific variance retained. Ensembl gene identifiers were mapped to HGNC gene symbols using the biomaRt package with the hsapiens_gene_ensembl dataset. Version numbers were removed from Ensembl IDs before annotation. Genes lacking HGNC symbols were excluded. If multiple Ensembl IDs mapped to a single HGNC symbol, their expression values were averaged during annotation. The final expression matrix contained VST-normalised values indexed by HGNC symbols and was saved for use in downstream analyses.

### 2.4. Curation of Ferroptosis-Related Genes

Ferroptosis-associated genes were obtained from FerrDb v2 (http://www.zhounan.org/ferrdb; accessed on 3 March 2025) [31]. Genes were extracted from four categories: drivers, suppressors, markers, and unclassified regulators. Genes lacking HGNC symbols were excluded. Each gene was annotated according to its functional role: drivers promote ferroptosis, suppressors inhibit it, markers are associated but not causative, and unclassified genes have undefined roles. Gene lists were merged and deduplicated using HGNC symbols. When genes were assigned to multiple categories, role annotations were combined.

### 2.5. Wilcoxon Analysis of Ferroptosis-Related Gene Expression

Differential expression of ferroptosis-related genes between LUAD tumours and normal tissues was assessed using a Wilcoxon rank-sum test. The analysis focused on the curated ferroptosis gene set (n = 1293). VST-normalised expression values were used as input, and only genes matched to the normalised matrix were retained. For each gene, expression differences between tumour (n = 539) and normal (n = 59) samples were compared. Log fold changes were calculated as the difference in median expression between the tumour and normal groups. *p*-values were adjusted using the Benjamini–Hochberg method. Genes with |logFC| > 1 and FDR < 0.05 were considered significantly differentially expressed. The top 30 significant ferroptosis-related genes were visualised using a heatmap of row-scaled VST expression values. Tumour and normal samples were annotated accordingly. The heatmap was generated using the pheatmap package [32].

### 2.6. Survival Analysis and Prognostic Gene Selection

Cox proportional hazards regression was applied to evaluate the prognostic relevance of ferroptosis-associated genes in LUAD. Expression values for the top 30 ferroptosis-related candidates, identified from the Wilcoxon analysis, were extracted from the VST-normalised expression matrix and merged with matched clinical metadata. Overall survival (OS) was defined as the number of days from diagnosis to death or last follow-up, with vital status coded as 1 (death) or 0 (alive). Univariate Cox regression models were fitted for each candidate gene using the survival package in R [33]. Genes demonstrating significant associations with OS (*p* < 0.05) were then jointly assessed in a multivariate Cox regression model to determine their independent prognostic contributions. Hazard ratios (HRs), 95% confidence intervals (CIs), and *p*-values were calculated. Model results were visualised using a forest plot. Based on the multivariate model, a prognostic gene signature was defined. Patient-specific risk scores were computed using the model coefficients and used to stratify patients into high- and low-risk groups. Kaplan–Meier survival curves with log-rank tests were used to compare OS between groups. Model performance was further assessed with time-dependent receiver operating characteristic (ROC) analysis at 1-, 3-, and 5-year intervals using the timeROC package [34]. Genes retained in the multivariate signature were carried forward for downstream pathway correlation and functional analyses.

### 2.7. Pathway Activity Scoring and Correlation with Prognostic Genes

To investigate biological pathways associated with the six prognostic ferroptosis-related genes, single-sample gene set enrichment analysis (ssGSEA) was performed. Eight hallmark pathways relevant to tumour progression and immune response were selected from the Molecular Signatures Database (MSigDB v2023.1) [35]. These included epithelial–mesenchymal transition, angiogenesis, hypoxia, inflammatory response, IL6-JAK-STAT3 signalling, TNFα-NFκB signalling, and interferon alpha and gamma responses. Pathway gene sets were retrieved using the msigdbr package in R and formatted for analysis. VST-normalised expression data were used as input for ssGSEA, implemented using the GSEApy Python package [36]. Normalised enrichment scores (NES) were calculated for each pathway across all LUAD samples. The ssGSEA matrix was exported for correlation analysis. Expression data for the six prognostic genes were matched to pathway activity scores. Spearman correlation coefficients and associated *p*-values were calculated to assess associations between gene expression and pathway activity, and *p*-values were Benjamini–Hochberg adjusted across all gene-pathway tests (6 × 8 comparisons). Correlation results were compiled into coefficient and *p*-value matrices. A clustered heatmap of correlation coefficients was generated to visualise gene-pathway associations.

### 2.8. Functional Enrichment Analysis

Functional enrichment analysis was performed to explore the biological roles of the six prognostic ferroptosis-related genes identified by multivariate Cox regression. The analysis was conducted using g:Profiler (https://biit.cs.ut.ee/gprofiler; accessed on 3 March 2025) [37], using HGNC gene symbols as input. The enrichment analysis included Gene Ontology categories (biological process, molecular function, cellular component), KEGG pathways, and Reactome pathways. The background set was defined as all human protein-coding genes available in the relevant Ensembl release. Multiple testing correction was applied using the g:SCS method. Enrichment terms with an adjusted *p*-value < 0.05 were considered significant. Top-ranking GO terms and pathways were used to interpret the potential biological functions of the prognostic genes.

### 2.9. Machine Learning Classification Using Six-Gene Signature

To assess the diagnostic potential of the identified six-gene panel, supervised machine learning models were trained to distinguish LUAD tumour samples from adjacent normal tissues. VST-normalised expression values for the six prognostic genes (AQP4, CDCA3, HJURP, KIF20A, PLK1, UHRF1) were extracted from the expression matrix and combined with TCGA sample metadata. Samples were randomly stratified into training (80%) and independent test (20%) sets. To mitigate class imbalance between tumour and normal tissues, the Synthetic Minority Over-sampling Technique (SMOTE) was applied during training [38]. Ten classification algorithms were evaluated: random forest (RF), support vector machine (SVM), extreme gradient boosting (XGB), adaptive boosting (ADA), gradient boosting machine (GBM), k-nearest neighbours (KNN), logistic regression (LOGIT), decision tree (TREE), Gaussian naïve Bayes (NB), and artificial neural network (ANN). Models were implemented using the scikit-learn (v1.3.2), xgboost (v1.7.6), and imbalanced-learn (v0.11.0) libraries in Python (v3.10). Hyperparameters were set to scikit-learn defaults unless otherwise specified (ANN: hidden layer size = 50, max_iter = 1000; logistic regression: max_iter = 1000). Model performance was evaluated using stratified five-fold cross-validation on the training set and subsequently tested on the independent hold-out set. Receiver operating characteristic (ROC) curves were generated for each model, and the area under the curve (AUC) was reported as the primary performance metric. In addition, accuracy, precision, recall, specificity, and F1-score were calculated to provide a comprehensive evaluation of classifier performance. ROC curves and performance heatmaps were generated using matplotlib (v3.8.2) and seaborn (v0.13.1).

### 2.10. Targeted Drug Prediction and Gene–Drug Correlation Analysis

To identify potential therapeutic compounds associated with the six prognostic ferroptosis-related genes, gene-drug correlation analysis was performed using the CellMiner database (https://discover.nci.nih.gov/cellminer; 3 March 2025) [39]. Drug sensitivity data (Z-scores) and RNA-seq-based gene expression profiles from the NCI-60 cancer cell line panel were obtained from CellMiner (version 2.14.1). For each gene, Pearson correlation analysis was conducted to assess associations between gene expression and drug activity across cell lines. Correlation coefficients (r) and *p*-values were generated using the Pattern Comparison tool in CellMiner. The top five compounds per gene were selected based on statistical significance (*p* < 0.01) and correlation strength (|r| ≥ 0.3). Positive correlations indicated that higher gene expression was associated with reduced drug sensitivity or potential resistance. In contrast, negative correlations suggested increased sensitivity. In CellMiner, more negative activity Z-scores typically indicate greater growth inhibition (higher sensitivity); accordingly, negative gene-drug correlations imply greater sensitivity with higher gene expression. Given the pan-cancer composition of the NCI-60 panel, these correlations may not reflect LUAD-specific biology and should be interpreted as exploratory. Gene-drug associations were visualised using scatter plots of gene expression versus drug activity scores. The overall workflow is summarized in Figure 1.

## 3. Results

### 3.1. Transcriptomic Landscape of LUAD Reveals Widespread Dysregulation

To delineate the transcriptional landscape of lung adenocarcinoma (LUAD), we analysed bulk RNA-sequencing data from The Cancer Genome Atlas (TCGA), comprising 539 primary tumours and 59 adjacent normal lung tissues. Following quality control and integration, a unified expression matrix was generated for 598 samples with harmonised gene and clinical annotations. Differential expression analysis performed in an unpaired design (all tumours vs. all normals) identified 4686 significantly dysregulated genes (Benjamini–Hochberg adjusted *p* < 0.05 and |log_2_FC| > 1), including 2597 upregulated and 2089 downregulated transcripts. The volcano plot highlighted strong upregulation of canonical oncogenic drivers such as FAM83A, TOP2A, and B3GNT3, alongside downregulation of signalling and structural genes including PECAM1 and EPAS1 (Figure 2A). Hierarchical heatmap analysis of the top 30 DEGs demonstrated clear segregation of tumour and normal samples, consistent with extensive transcriptional reprogramming (Appendix A). To account for patient-specific baseline variability, we next performed a paired analysis restricted to 58 patients with matched tumour-normal samples. Although the total number of DEGs was reduced compared with the unpaired design, the analysis retained strong contrasts between conditions. Notably, FAM83A and FABP4 emerged as consistently upregulated across patients, while ITLN2 and SLC6A4 were among the most downregulated (Figure 2B and Appendix A). Clustering of the top 30 paired DEGs again revealed robust separation of matched tumour and normal tissues. Together, these complementary analyses demonstrate that LUAD is characterised by robust, patient-consistent transcriptional reprogramming, with key markers such as FAM83A, FABP4, and ITLN2 remaining significant even under the more stringent paired design. Full differential expression statistics are provided in Appendix A.

### 3.2. Normalisation and Ferroptosis Gene Annotation Reveal a Refined Expression Matrix for LUAD

To ensure comparability across LUAD samples and account for variability in sequencing depth, raw gene expression counts were subjected to variance-stabilising transformation (VST), producing log-scaled, homoscedastic values suitable for downstream statistical modelling. Ensembl gene identifiers were mapped to HGNC symbols to enhance biological interpretability, and entries lacking valid gene symbols were excluded. Where multiple Ensembl IDs corresponded to the same HGNC gene, expression values were averaged. The resulting dataset comprised VST-normalised expression values for uniquely annotated genes, forming the foundation for functional and clinical analyses. In parallel, we curated a comprehensive list of ferroptosis-associated genes from FerrDb v2, encompassing experimentally supported drivers, suppressors, markers, and unclassified regulators. After rigorous filtering and annotation, including exclusion of genes lacking HGNC symbols and consolidation of role assignments for duplicates, we assembled a reference set of 1293 unique ferroptosis-related genes (Appendix A). This catalogue reflects the current functional landscape of ferroptosis in human biology and provides a reference framework for subsequent expression and survival analyses in LUAD. This harmonised dataset ensures analytical robustness and provides a biologically anchored framework for investigating the role of ferroptosis in LUAD.

### 3.3. Dysregulation of Ferroptosis-Related Genes in LUAD Tumours

To assess the relevance of ferroptosis in LUAD pathology, we intersected our differentially expressed gene (DEG) list with the curated ferroptosis gene set. From the 1293 ferroptosis-associated genes, 182 were significantly dysregulated in LUAD tumours compared to adjacent normal lung tissue (adjusted *p* < 0.05, |log_2_FC| > 1) (Figure 3). These included representatives from all functional categories, reflecting widespread perturbation of ferroptosis regulation in the LUAD transcriptome. A heatmap of the top 30 differentially expressed ferroptosis genes, ranked by statistical significance, revealed a clear separation between tumour and normal samples (Appendix A). This subset highlighted distinct transcriptional signatures potentially linked to ferroptotic processes and redox imbalance in LUAD. The enrichment of ferroptosis-related genes among DEGs highlights their clinical and biological importance in LUAD, justifying their prioritisation in prognostic modelling. A full list of intersecting genes is provided in Appendix A, with differential expression values for all significant genes in Appendix A, and ranked statistics for the top 30 genes in Appendix A.

### 3.4. A Ferroptosis-Derived Signature Stratifies LUAD Patients by Survival Risk

To determine whether ferroptosis-associated genes carry prognostic information in LUAD, we first applied univariate Cox proportional hazards regression across the 30 candidates selected from the Wilcoxon test (Methods). Six genes, AQP4, CDCA3, HJURP, KIF20A, PLK1 and UHRF1, showed significant associations with overall survival (OS; *p* < 0.05; Appendix A). These six were then evaluated together in a multivariable Cox model to identify independent effects. Of the panel, PLK1 remained significant (HR = 1.46, 95% CI = 1.09–1.94, p = 0.01), consistent with its central role in mitotic regulation, whereas the other genes did not retain independent predictive value once correlations were accounted for (Figure 4). Using the regression coefficients from this model, we derived a composite risk score for each patient. Stratification at the median risk score separated the cohort into high- and low-risk groups with markedly divergent outcomes: patients in the high-risk group had significantly shorter OS compared to those in the low-risk group (log-rank *p* < 0.0001; Figure 5A). Time-dependent ROC analyses confirmed the discriminatory capacity of the model, yielding AUC values of 0.83, 0.76, and 0.71 at 1, 3, and 5 years, respectively (Figure 5B–D). Taken together, these results establish a ferroptosis-derived gene set that robustly stratifies LUAD patients by survival risk, despite redundancy among individual members. While the prognostic signal is dominated by PLK1, the integrated risk model provides clinically relevant stratification beyond single-gene associations. This finding emphasises PLK1 as a dominant ferroptosis-linked driver of LUAD prognosis, consistent with its central role in cell-cycle regulation. Patient-level risk scores are provided in Appendix A. To gain mechanistic insights into how these prognostic genes may shape LUAD biology, we next examined their relationship to hallmark tumour pathways.

To evaluate whether the six-gene ferroptosis panel provides prognostic information beyond standard clinical parameters, we integrated the RiskScore with AJCC pathological stage, age, and sex in multivariable Cox regression models (Appendix A). In the stage-only model, a higher stage was significantly associated with poorer overall survival (HR = 1.65, 95% CI 1.39–1.96, p = 1.8 × 10^−8^; concordance = 0.682). When modelled independently, the six-gene RiskScore was strongly prognostic (HR = 3.44, 95% CI 2.16–5.48, p = 1.9 × 10^−7^; concordance = 0.635). In the combined model, stage and RiskScore both retained independent significance, with RiskScore conferring nearly a three-fold increased hazard (HR = 2.99, 95% CI 1.80–4.97, p = 2.3 × 10^−5^). Age and sex were not significant predictors.

Discriminative performance improved when the RiskScore was added to stage. The concordance index increased from 0.682 (stage ± age/sex) to 0.703 in the combined model. A likelihood-ratio test confirmed that including the RiskScore significantly improved model fit beyond stage alone (χ^2^ = 14.99, df = 1, p = 1.1 × 10^−4^). Out-of-fold time-dependent ROC analyses further supported incremental prognostic utility: AUCs for the combined model were 0.710, 0.754, and 0.730 at 1, 3, and 5 years, respectively, compared with 0.680, 0.701, and 0.721 for stage alone (Appendix A, panels C,D; Appendix A). Kaplan–Meier analysis stratified by median RiskScore showed significantly worse survival for high-risk patients (log-rank *p* = 3.8 × 10^−4^; Appendix A, panel B). Together, these results demonstrate that the six-gene RiskScore provides robust, independent prognostic information that complements and enhances traditional staging. This underscores the potential of the six-gene RiskScore as a clinically relevant adjunct to staging systems in LUAD.

### 3.5. Prognostic Ferroptosis Genes Associate with LUAD Hallmark Pathways

To investigate the biological context of the six prognostic genes, we first evaluated global and gene-restricted transcriptional programs. Principal component analysis (PCA) of hallmark pathway activity scores derived from ssGSEA revealed a clear separation between LUAD and normal tissues along the first two principal components, underscoring systematic differences in pathway activity between tumour and control samples (Figure 6A). Notably, when PCA was repeated using only the six prognostic genes (AQP4, CDCA3, HJURP, KIF20A, PLK1, UHRF1), tumour and normal samples remained partially separable (Figure 6B). This indicates that despite redundancy among these genes, the panel captures key expression features reflective of LUAD-associated transcriptional reprogramming. We next quantified pathway-level activity for eight tumour-relevant processes, including epithelial–mesenchymal transition (EMT), angiogenesis, hypoxia, inflammatory response, IL6-JAK-STAT3 signalling, TNFα-NFκB signalling, and interferon responses (Methods). LUAD samples demonstrated widespread enrichment across these pathways, consistent with the activation of pro-tumourigenic and immune-modulatory signalling programs (Appendix A).

Correlation analysis between the six prognostic genes and pathway activity scores revealed distinct association patterns (Appendix A). AQP4 expression showed significant positive correlations with angiogenesis (ρ = 0.12, FDR adj. *p* = 0.004) and EMT (ρ = 0.16, FDR adj. *p* = 0.0001), suggesting its involvement in the microenvironmental remodelling. By contrast, PLK1, CDCA3, and HJURP exhibited weak but consistent negative correlations with EMT, in line with their established roles in cell-cycle regulation rather than mesenchymal programs. Network analysis of significant gene–pathway correlations further highlighted these divergent patterns, with AQP4 linked to angiogenesis and EMT, while proliferation-associated genes clustered with negative edges towards EMT (Figure 6C). Expression comparisons confirmed that all six prognostic genes were significantly dysregulated between LUAD and normal samples (Figure 6D). A heatmap and barplot summarising these associations highlight that while PLK1 dominates prognostic modelling, AQP4 uniquely links the signature to angiogenic and mesenchymal signalling (Figure 7A,B). Thus, the panel integrates both proliferation-driven and microenvironmental signals, which may explain its strong prognostic performance.

### 3.6. Prognostic Ferroptosis Genes Converge on Cell Cycle and Genomic Integrity Pathways

To elucidate the biological processes underpinning the six prognostic genes, we performed functional enrichment analysis using g:Profiler. Despite their initial nomination from a ferroptosis-related catalogue, enrichment was strongly dominated by cell cycle–linked terms (Appendix A). Significant Gene Ontology (GO) and Reactome categories included spindle organisation (GO:0005819, adjusted *p* = 0.027) and Mitotic Telophase/Cytokinesis (REAC:R-HSA-68884, adjusted *p* = 0.0026), alongside a molecular function term for histone H3 ubiquitin ligase activity (GO:0141055, adjusted *p* = 0.049) largely attributable to UHRF1. Collectively, these terms converge on processes critical for genomic integrity, proliferative control, and chromosomal segregation, hallmarks of aggressive LUAD biology. Visualisation of enriched categories in a dot plot (Figure 8) highlighted recurrent involvement of KIF20A and PLK1 in mitotic spindle and cytokinesis regulation, consistent with their known roles as proliferation drivers. The prominence of these pathways indicates that the apparent “ferroptosis-linked” signature is mechanistically anchored in cell cycle regulation, with genomic stability checkpoints (G2/M transition, chromosome segregation) emerging as key axes of association. These findings align with reviewer concerns by reframing the panel not as ferroptosis-specific but as a proliferation-centric signature that nonetheless retains robust prognostic value. This suggests that although seeded from ferroptosis, the panel captures proliferation pathways that are tightly linked to LUAD aggressiveness and outcome.

### 3.7. Six-Gene Proliferation-Linked Signature Discriminates LUAD from Normal Tissue Across Machine Learning Models

To assess whether the six-gene panel could distinguish LUAD from normal lung tissue, we trained ten supervised machine learning classifiers on VST-normalised expression profiles. Models included random forest (RF), support vector machine (SVM), extreme gradient boosting (XGB), adaptive boosting (ADA), gradient boosting machine (GBM), k-nearest neighbours (KNN), logistic regression (LOGIT), decision tree (TREE), naive Bayes (NB), and artificial neural network (ANN). Model performance was evaluated using stratified five-fold cross-validation within the training set and independently tested on a held-out set, thereby separating model optimisation from evaluation. Receiver operating characteristic (ROC) analyses demonstrated that most classifiers achieved near-perfect separation of LUAD and normal samples (Figure 9A). RF, SVM, XGB, ADA, GBM, NB, and ANN each reached an AUC of 1.00, while KNN (AUC = 0.99) showed slightly reduced performance, whereas LOGIT also reached AUC = 1.00; TREE performed lowest (AUC = 0.96). Beyond AUC, models were benchmarked across accuracy, precision, recall, specificity, and F1-score (Figure 9B; Appendix A). Ensemble methods (RF, GBM, XGB) consistently achieved perfect scores across all metrics, confirming their robustness, whereas simpler models such as TREE and KNN displayed minor reductions in recall and specificity. The uniformly high performance highlights that expression differences between LUAD and adjacent normal tissues are sufficiently pronounced that even small gene panels can achieve strong discriminatory accuracy. Nevertheless, these results should be interpreted with caution. Given the substantial sample imbalance (539 tumours vs. 59 normals) and the stark biological differences between malignant and non-malignant lung tissue, the classification task is intrinsically easier than distinguishing clinically challenging scenarios (e.g., benign nodules vs. early-stage LUAD). Although SMOTE balancing and cross-validation reduced the risk of bias, the possibility of overfitting cannot be excluded, and the absence of validation in independent LUAD cohorts remains a limitation. Together, these findings indicate that the six-gene panel reliably separates LUAD from normal samples across diverse algorithms, reinforcing its potential diagnostic utility in principle. However, its clinical application will require evaluation in external cohorts and more relevant contexts (small biopsies or liquid biopsy assays) before it can be considered for translational use.

### 3.8. Gene-Drug Correlations Highlight Candidate Vulnerabilities but Require LUAD-Specific Validation

To explore whether the six-gene panel might inform therapeutic sensitivity, we queried the CellMiner platform, which integrates gene expression and compound activity profiles across the NCI-60 cancer cell line panel. Pearson correlation analysis identified 30 statistically significant gene–drug associations (|r| ≥ 0.3, *p* < 0.01), spanning all six prognostic genes (Appendix A). Several correlations suggested potential vulnerabilities in proliferative LUAD contexts. For instance, AQP4 expression was negatively correlated with multiple investigational agents, including STK527948 (NSC663954, r ≈ −0.64) and NSC831100 (r ≈ −0.72, *p* < 0.001), indicating that higher AQP4 expression may sensitise cells to these compounds. Similarly, CDCA3 and HJURP displayed reproducible negative correlations with distinct NSC derivatives (e.g., NSC685416, NSC709900), consistent with enhanced drug sensitivity in tumours with elevated expression. In contrast, UHRF1 exhibited positive correlations with compounds such as NSC709319, suggesting a possible resistance phenotype, while KIF20A and PLK1 showed negative associations with small molecules, including NSC715595 and NSC690672. Collectively, these correlations implicate proliferation-linked genes in shaping compound response, pointing to candidate drug sensitivities that warrant further investigation.

It is important to interpret these results with caution. The NCI-60 panel comprises diverse tumour types, and correlations derived from this mixed background may not fully capture LUAD-specific biology. Moreover, most compounds identified are preclinical and lack well-characterised mechanisms of action. Notably, PLK1, the sole independent prognostic gene in our multivariate analysis, is already the focus of active drug development, with inhibitors such as volasertib and onvansertib being evaluated in oncology trials. This highlights a more direct translational avenue: high PLK1 expression could serve as a biomarker for patient stratification in PLK1-targeted therapies. By contrast, the associations for other genes, particularly AQP4, remain exploratory and require careful reconciliation with prior reports showing conflicting prognostic roles. Together, these findings nominate a set of candidate compounds linked to expression of the six-gene panel, providing a foundation for hypothesis generation. However, rigorous validation in LUAD-specific preclinical models will be essential to determine whether these associations translate into actionable therapeutic strategies. These exploratory correlations highlight possible therapeutic vulnerabilities, particularly for PLK1, that may guide future biomarker-driven LUAD trials. Because individual scatterplots are hard to parse at scale, we present the gene–drug associations primarily as Appendix A (top five compounds per gene, with r and *p* values); Appendix A is retained only as a compact visual summary.

## 4. Discussion

Lung adenocarcinoma (LUAD) remains a formidable clinical challenge, representing the most prevalent subtype of non-small-cell lung cancer and contributing substantially to global cancer mortality [40,41]. Despite advances in targeted therapies and immune checkpoint inhibitors, only a subset of patients derive a durable benefit due to intertumoural heterogeneity and the absence of actionable mutations in many cases [42]. This underscores a pressing need for novel diagnostic and prognostic biomarkers capable of identifying high-risk patients and guiding individualised therapy. Ferroptosis, a non-apoptotic form of iron-dependent cell death, has emerged as a promising but underexplored mechanism in cancer pathobiology. Although extensively studied in experimental systems, the translational relevance of ferroptosis in LUAD remains largely undefined [43,44,45]. In this context, our study presents integrative evidence for a six-gene expression signature related to ferroptosis, comprising AQP4, CDCA3, HJURP, KIF20A, PLK1, and UHRF1, with robust diagnostic and prognostic utility in LUAD, and highlights its potential as a foundation for precision therapeutics.

The proposed six-gene panel addresses a critical diagnostic gap in LUAD by extending beyond the conventional mutation-centric paradigm. While EGFR, ALK, and KRAS alterations are key biomarkers for stratified therapies, a large proportion of LUAD patients remain genetically unclassifiable and thus lack targeted treatment options [40,41]. Gene expression-based classifiers, particularly those reflecting dynamic cellular states such as oxidative stress and lipid peroxidation, offer a functional layer of stratification. Ferroptosis sits at the intersection of these processes [46], making it a biologically and clinically relevant lens through which to interrogate LUAD. Prior investigations into ferroptosis in lung cancer have largely focused on regulators like GPX4 or SLC7A11 [47]; in contrast, our work captures a broader ferroptosis-linked transcriptomic signature with direct prognostic implications. Notably, several members of the panel, PLK1, UHRF1, and HJURP, have known roles in mitotic regulation and chromatin remodelling, both of which intersect with ferroptotic sensitivity and tumour aggressiveness [44,45]. Less characterised genes like AQP4 and CDCA3 showed consistent dysregulation in LUAD and merit further exploration. Collectively, this gene panel stratifies patients into high- and low-risk categories with clear survival disparities, offering an accessible, mutation-independent tool for clinical decision-making.

Crucially, this gene signature is more than a statistical classifier; it reflects core oncogenic programs in LUAD. Our functional analyses revealed strong associations between panel genes and cancer hallmarks, including epithelial–mesenchymal transition (EMT), angiogenesis, hypoxia, and inflammatory signalling pathways intimately linked to tumour progression and immune evasion [40,41]. AQP4, for example, showed positive correlations with EMT and angiogenesis scores, consistent with its proposed role in tumour invasion and vascular permeability [42]. Conversely, PLK1 and CDCA3 were inversely associated with EMT but tightly linked to cell cycle progression, highlighting a mechanistic divergence within the panel [43]. Enrichment analyses pointed to DNA replication, G2/M checkpoints, and chromosomal segregation as dominant pathways, a finding that aligns with the mitotic stress vulnerabilities commonly observed in LUAD. Of particular note, PLK1 and UHRF1 emerged as potential oncogenic hubs, reaffirming their roles in both tumour proliferation and ferroptotic resistance [45,48]. These data collectively position the six-gene panel not only as a classifier but also as a biological fingerprint of LUAD aggressiveness and therapeutic liability.

Current diagnostic strategies for LUAD focus primarily on histopathology and the detection of specific driver mutations like EGFR, ALK, and ROS1 [40]. While effective for a subset of patients, this approach leaves many without clear diagnostic or therapeutic guidance. In contrast, gene expression-based models offer a broader view of tumour biology and can support diagnosis even in mutation-negative cases [41]. In contrast, gene expression-based models offer a broader view of tumour biology and can support diagnosis even in mutation-negative cases [49] and are readily translatable using routine expression assays. Prognostically, the gene panel provided clear stratification of survival outcomes between low- and high-risk groups, with the risk score performing independently of common clinical covariates. Importantly, integration with AJCC stage improved discriminative accuracy and model fit, underscoring the incremental prognostic value of the panel over conventional staging. Taken together, these findings elevate the panel’s clinical utility: it is compact, interpretable, and compatible with standard workflows, which may facilitate implementation even in resource-limited settings.

Beyond diagnosis and prognosis, we also explored therapeutic implications by linking gene expression profiles with compound sensitivity. Using CellMiner’s integrated drug response datasets, we uncovered gene–drug associations that could inform off-label or repurposed treatment strategies. High AQP4 expression was negatively correlated with several investigational agents, including STK527948 (NSC663954), suggesting potential sensitivity in AQP4-overexpressing LUAD tumours. Similarly, UHRF1 showed strong associations with experimental thiadiazine and benzothiazinone derivatives (e.g., NSC709319, NSC635132), implicating this epigenetic regulator as a candidate predictor of response to chromatin-targeting compounds. In contrast, PLK1 expression correlated with several small molecules (e.g., NSC690672, NSC696861), yet the direction and magnitude of response suggest possible resistance mechanisms that require mechanistic clarification. While most of the drugs identified remain preclinical and lack clinical annotation, the observed expression–activity correlations lay the groundwork for precision-guided preclinical testing. These findings support a shift from mutation-guided to expression-driven therapeutic stratification in LUAD. However, because the NCI-60 dataset represents a pan-cancer background, these results must be considered exploratory until validated in LUAD-specific preclinical systems.

Looking forward, several avenues remain to fully unlock the translational value of this signature. External validation in independent LUAD cohorts, especially those with diverse ethnic and clinical backgrounds, is essential for confirming its robustness and generalisability. Functional studies in LUAD models, including patient-derived organoids and xenografts, will help delineate the mechanistic contributions of AQP4, CDCA3, and other panel members in ferroptotic regulation. The observed drug–gene correlations provide a pipeline for preclinical drug screening and could inform prospective trials focused on biomarker-guided therapy. Finally, integration of this expression-based panel into multimodal diagnostic platforms alongside radiomics, proteomics, and other modalities may further enhance predictive power and clinical utility.

This study delivers a ferroptosis-linked six-gene expression signature that bridges diagnostic accuracy, prognostic value, and therapeutic insight in LUAD. It complements and extends existing genomic classifiers, addresses critical gaps in patient stratification, and lays the groundwork for biomarker-informed therapeutic strategies. With further clinical validation, this panel has the potential to advance personalised medicine in LUAD, improving outcomes in a patient population that remains underserved by current molecular tools.

## 5. Conclusions

Ferroptosis is a distinct form of regulated cell death with increasing relevance in oncology, yet its role in lung adenocarcinoma (LUAD) remains underexplored. In this study, we identified a clinically relevant six-gene expression signature (AQP4, CDCA3, HJURP, KIF20A, PLK1, and UHRF1) linked to ferroptosis that supports diagnosis, prognosis, and therapeutic insight in LUAD. Derived from transcriptomic analysis of TCGA cohorts and evaluated through machine learning and drug sensitivity correlations, the panel effectively distinguishes tumour from normal tissue, stratifies patient survival risk, and suggests potential therapeutic vulnerabilities. Compared to mutation-based biomarkers, this expression-based signature captures real-time tumour biology, including oxidative stress, chromatin remodelling, and immune modulation. Its compact size and consistent performance support its potential for clinical application, particularly in patients lacking canonical driver mutations. Together, our findings establish a ferroptosis-linked gene signature with translational relevance in LUAD, offering a practical framework to support diagnosis, prognosis, and therapy selection in a disease with significant unmet clinical needs.

## 6. Limitations

However, several limitations should be noted. First, external validation was not performed due to heterogeneity among public datasets in terms of platform, normalisation, and lack of sufficient normal controls. Second, drug–gene correlations were based on the NCI-60 panel, which is pan-cancer and not LUAD-specific. Third, the study is computational in nature, and experimental validation—such as ferroptosis assays, gene knockdown, or drug response studies—is needed to confirm the findings. Lastly, the imbalance between tumour and normal samples in TCGA (539 vs. 59) may influence differential expression analysis despite statistical adjustments. Future work should include validation in independent LUAD cohorts with harmonised sequencing and robust experimental testing to confirm the prognostic and functional roles of this panel.

## Figures and Tables

**Figure 1 biology-14-01280-f001:**
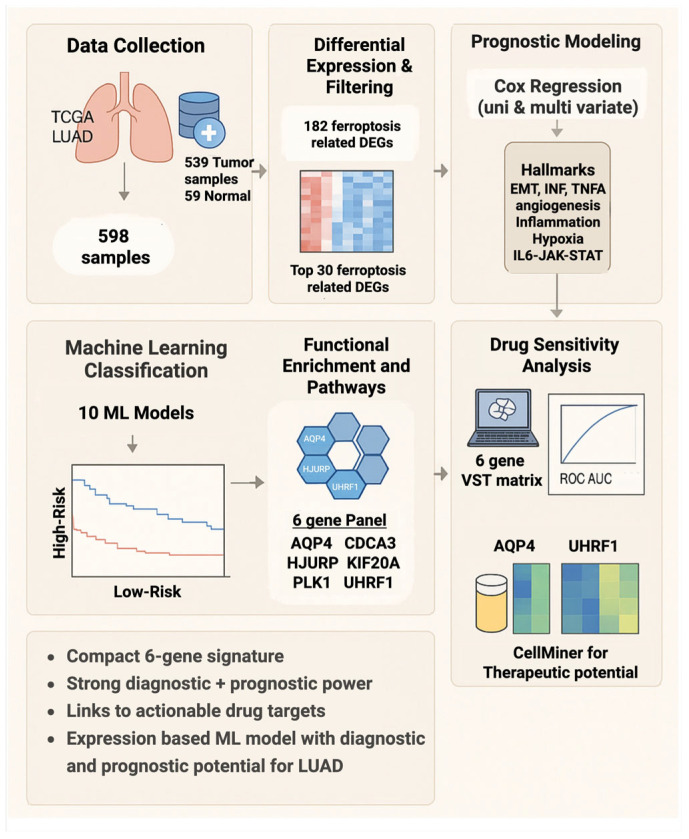
Workflow overview. TCGA LUAD RNA-seq data were analysed for differentially expressed genes, intersected with ferroptosis-related genes, and filtered by survival analysis to identify a six-gene panel. This panel was assessed for prognostic and diagnostic value, functional associations, and drug sensitivity correlations.

**Figure 2 biology-14-01280-f002:**
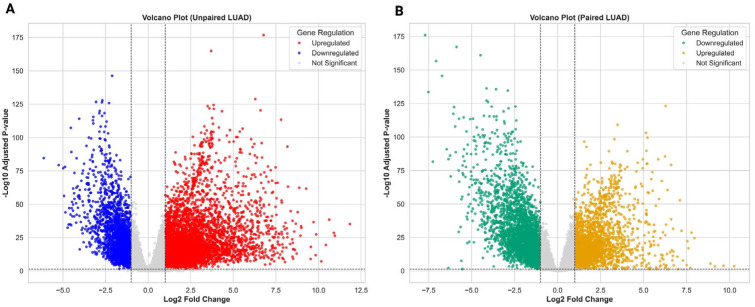
Transcriptomic dysregulation in LUAD. (**A**) Volcano plot showing differentially expressed genes (DEGs) from the unpaired analysis (539 tumours vs. 59 normal tissues). Genes with adjusted *p* < 0.05 and |log_2_FC| > 1 are classified as upregulated (red) or downregulated (blue). (**B**) Volcano plot from the paired analysis (58 matched tumour-normal pairs), highlighting fewer but patient-consistent DEGs.

**Figure 3 biology-14-01280-f003:**
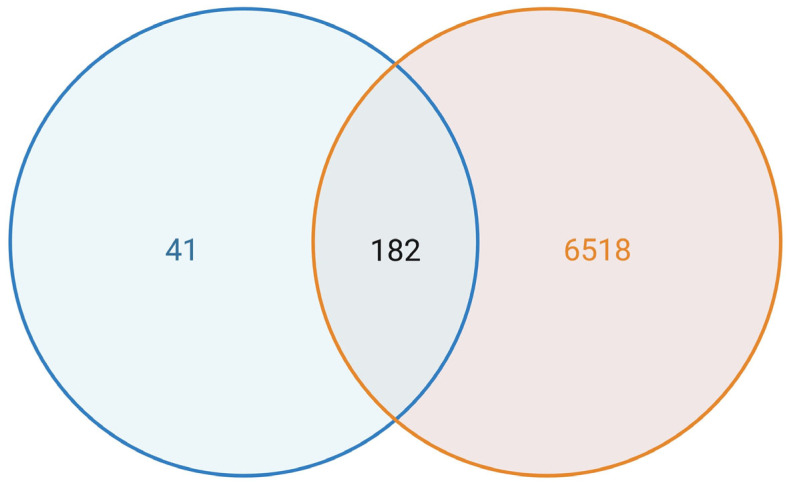
Ferroptosis-related gene expression in LUAD. Venn diagram showing the overlap between differentially expressed genes in LUAD and the curated ferroptosis gene set. A total of 182 ferroptosis-related genes were identified as differentially expressed in tumour samples.

**Figure 4 biology-14-01280-f004:**
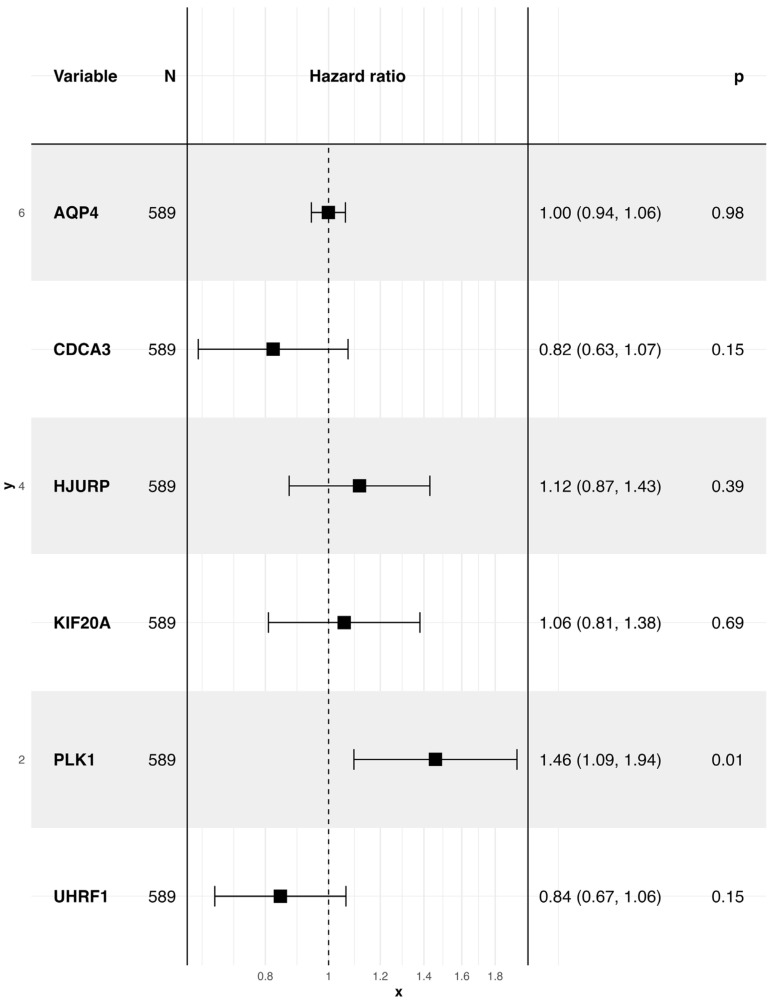
Multivariate Cox regression analysis of ferroptosis-related genes. Forest plot showing hazard ratios and 95% confidence intervals for genes included in the multivariate Cox model. *PLK1* was identified as an independent prognostic gene.

**Figure 5 biology-14-01280-f005:**
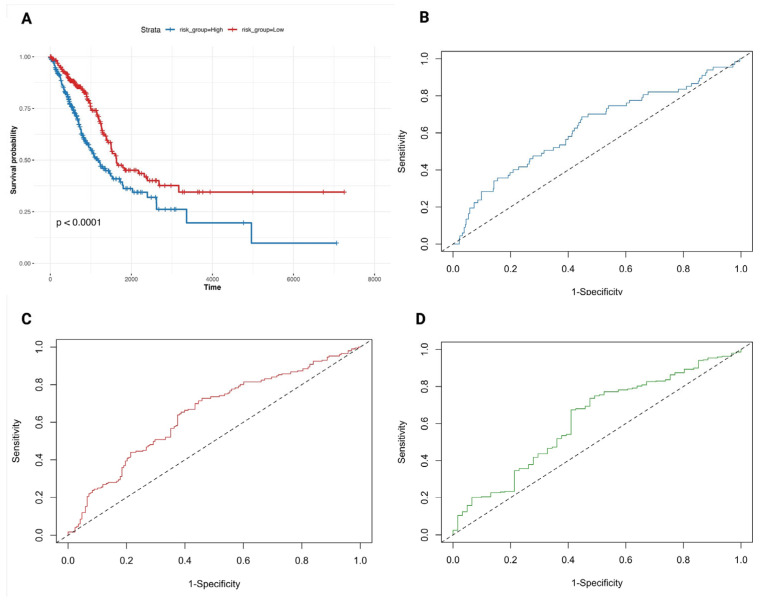
Prognostic value of the ferroptosis-related risk model in LUAD. (**A**) Kaplan–Meier survival curve comparing overall survival between high- and low-risk groups. (**B**–**D**) Time-dependent ROC curves showing model sensitivity and specificity at 1 year ((**B**), blue), 3 years ((**C**), red), and 5 years ((**D**), green).

**Figure 6 biology-14-01280-f006:**
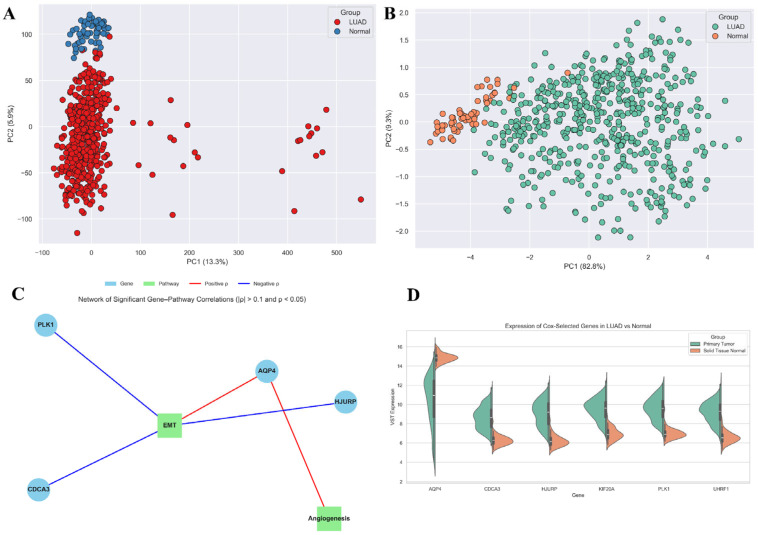
Pathway activity and expression of prognostic genes in LUAD. (**A**) PCA of ssGSEA hallmark pathway scores shows clear separation between LUAD tumours (red) and normal tissues (blue). (**B**) PCA using the six prognostic genes (AQP4, CDCA3, HJURP, KIF20A, PLK1, UHRF1) shows partial separation. (**C**) Network of significant gene–pathway correlations (|ρ| > 0.1, adjusted *p* < 0.05). Red: positive; blue: negative correlations. (**D**) Violin plots showing expression differences in the six genes between tumour and normal tissues.

**Figure 7 biology-14-01280-f007:**
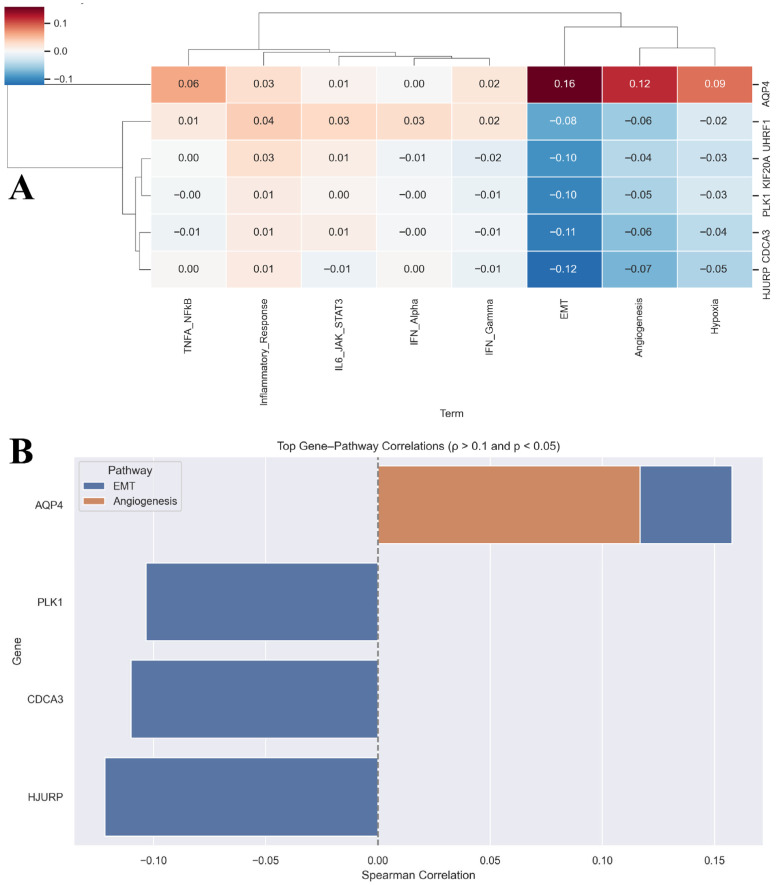
Correlation of prognostic genes with hallmark pathways in LUAD. (**A**) Heatmap of Spearman correlation coefficients between the six prognostic genes and ssGSEA hallmark pathway scores. Positive correlations are in red; negative in blue. (**B**) Bar plot summarising significant correlations (|ρ| > 0.1, adjusted *p* < 0.05). Bar length reflects correlation strength; pathways are colour-coded.

**Figure 8 biology-14-01280-f008:**
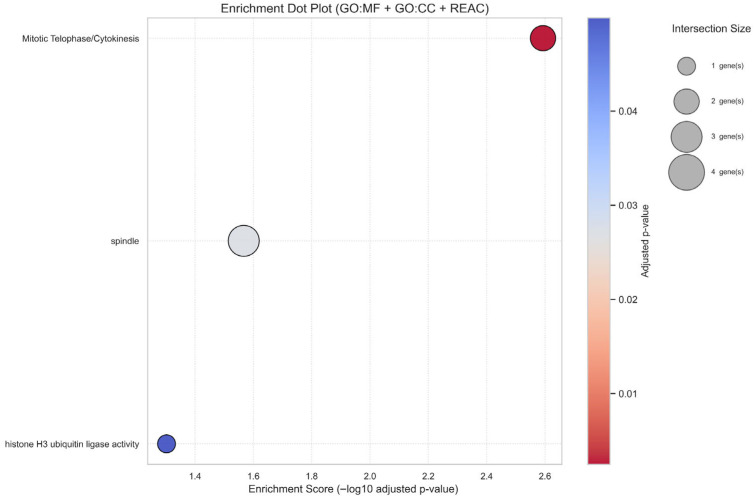
Enrichment analysis of prognostic genes. Dot plot showing significantly enriched GO (molecular function, cellular component) and Reactome terms for the six genes. Dot size reflects gene count; colour indicates adjusted *p*-value. Enriched categories highlight cell cycle and spindle-related processes, with UHRF1 linked to histone H3 ubiquitin ligase activity.

**Figure 9 biology-14-01280-f009:**
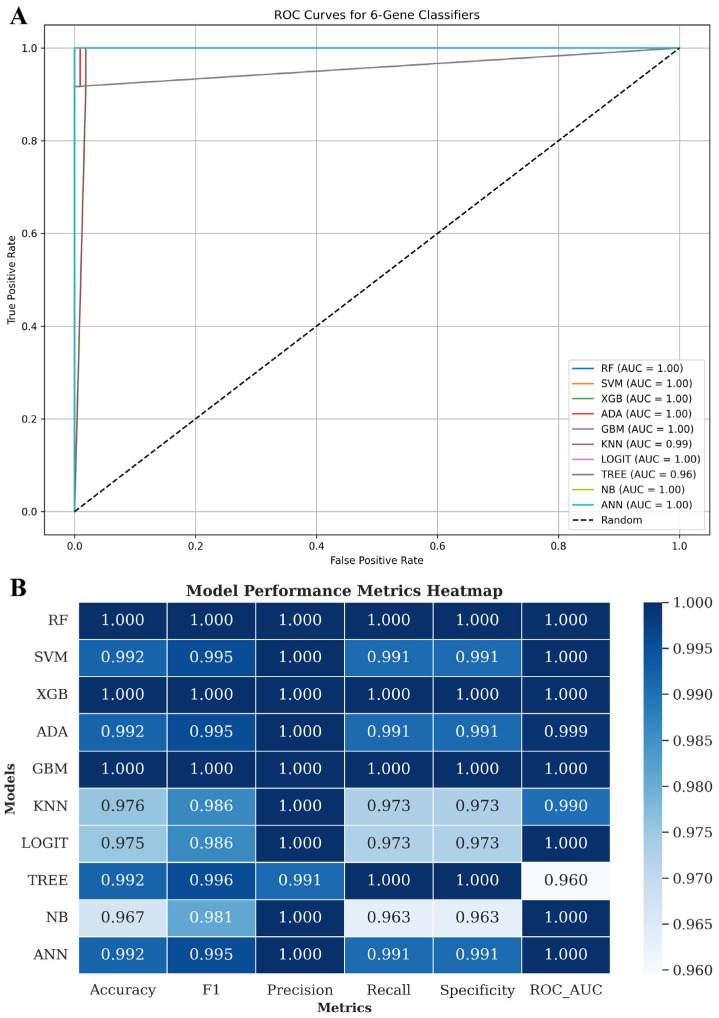
Machine learning evaluation of the six-gene LUAD classifier. (**A**) ROC curves showing test-set performance across ten machine learning models, with AUC values indicated. (**B**) Heatmap summarising test-set metrics: accuracy, precision, recall, specificity, F1-score, and ROC AUC.

## Data Availability

The RNA-sequencing and clinical data for lung adenocarcinoma (LUAD) used in this study are publicly available from The Cancer Genome Atlas (TCGA) via the Genomic Data Commons portal (https://portal.gdc.cancer.gov/). Drug sensitivity data and gene expression profiles from the NCI-60 cell line panel were obtained from the CellMiner database (version 2.14.1; https://discover.nci.nih.gov/cellminer; accessed on 3 March 2025). All processed datasets, including the six-gene expression matrix, risk scores, and gene–drug correlation results, are available in the Appendix A. Custom R and Python scripts used for analysis are also available upon request.

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
