# Peer review of "Ferroptosis-Linked Six-Gene Panel Enables Machine Learning-Assisted Diagnosis and Therapeutic Guidance in Lung Adenocarcinoma"

_biology, 2025, doi:10.3390/biology14091280_

Round 1

Reviewer 1 Report

Comments and Suggestions for Authors

The authors present a manuscript on the identification of a gene signature associated with ferroptosis for potential diagnosis, prognosis, and therapy in lung adenocarcinoma. The study involves an integrated approach combining differential expression analysis of TCGA-based RNA-Seq data to identify key candidate genes, as well as a number of analytical approaches to assess their biological and potential clinical significance.
Despite the fairly good structure of the article, a number of methodological clarifications are required for its publication.

Major comments

1) The dataset used by the authors has a significant imbalance (tumor n = 539, normal n = 59). Was the differential gene expression analysis considered in a paired mode (tumor-adjacent normal from the same patient)? If information on paired analysis is available, paired DESeq2 analysis may provide more accurate results. 2) The authors need to include an independent LUAD cohort to assess the generalizability of the results, as all analyses were performed on TCGA data only. If no additional datasets are included, the authors need to highlight this as a limitation and discuss the potential risks of overfitting the models.
3) Section 3.7 is unclear as to whether the model metrics are related to the training or test datasets. This needs to be clarified in the text. Also, given the high AUC values, the possibility of overfitting should be considered even when balancing with SMOTE.

4) The CellMiner dataset contains data for all cancer types, so correlations obtained for mixed cancer types may not reflect LUAD-specific biology. The authors need to discuss this.

Minor comments

1) Gene names should be italicized throughout the manuscript, following standard gene nomenclature formatting.
2) In Figure 2B, samples should be clustered by group. Overall, the heatmap is completely uninformative, showing only strong expression noise between groups. Gene names are unclear. The authors should reconsider the need for it.
3) Similar comments for Figure 3B
4) Figure numbering errors on lines 400 and 437; this should be corrected
5) In the figure of drug-gene correlations (current Figure 2 in Section 3.8), the scatterplots look difficult to interpret. The authors should replace this figure with a table for informational purposes.
6) Consecutive literature references on lines 46, 58, and 64 should be combined into one grouped reference (e.g., [1-3] instead of [1][2][3]). Also, references appearing in the middle of a sentence should be moved to the end for readability. 7) In line 158, the reference is incorrectly formatted
8) In line 230, "FDR" should probably be used instead of "P, adjusted for Benjamini-Hochberg" for clarity.
9) In line 231, there is a color typo in "log2FC"

Author Response

Reviewer 1

Comments and Suggestions for Authors

The authors present a manuscript on the identification of a gene signature associated with ferroptosis for potential diagnosis, prognosis, and therapy in lung adenocarcinoma. The study involves an integrated approach combining differential expression analysis of TCGA-based RNA-Seq data to identify key candidate genes, as well as a number of analytical approaches to assess their biological and potential clinical significance.
Despite the fairly good structure of the article, a number of methodological clarifications are required for its publication.

Response: I sincerely thank Reviewer 1 for the constructive feedback, which has helped substantially improve the clarity and robustness of our manuscript.

Major comments

1) The dataset used by the authors has a significant imbalance (tumor n = 539, normal n = 59). Was the differential gene expression analysis considered in a paired mode (tumor-adjacent normal from the same patient)? If information on paired analysis is available, paired DESeq2 analysis may provide more accurate results.

Response: Paired analysis: I agree with the reviewer. In addition to the unpaired DESeq2 analysis, I have now performed a paired analysis restricted to 58 LUAD tumour–normal pairs. The results confirmed consistent differential expression (new Fig. 2B; Supplementary Fig. S1B). These changes are described in Section 3.1.

2) The authors need to include an independent LUAD cohort to assess the generalizability of the results, as all analyses were performed on TCGA data only. If no additional datasets are included, the authors need to highlight this as a limitation and discuss the potential risks of overfitting the models.

Response: Independent LUAD cohort: I appreciate this important point. While surveying GEO and other repositories, dataset heterogeneity and a lack of sufficient normal samples limited integration. We now explicitly emphasize this as a key limitation in the Limitations section.

3) Section 3.7 is unclear as to whether the model metrics are related to the training or test datasets. This needs to be clarified in the text. Also, given the high AUC values, the possibility of overfitting should be considered even when balancing with SMOTE.

Response: Section 3.7 – training vs test metrics: I clarified that all model metrics reported are based on independent test sets, with training conducted using stratified five-fold cross-validation. I also added a cautionary note on potential overfitting despite SMOTE balancing.

4) The CellMiner dataset contains data for all cancer types, so correlations obtained for mixed cancer types may not reflect LUAD-specific biology. The authors need to discuss this.

Response: CellMiner pan-cancer dataset: I agree with the reviewer. The Results (Section 3.8) and Limitations (Section 5) now highlight that drug-gene correlations are exploratory due to the pan-cancer composition of NCI-60.

Minor comments

  • Gene names should be italicized throughout the manuscript, following standard gene nomenclature formatting.

Response: Gene names have been italicised throughout the manuscript following HGNC guidelines.

  • In Figure 2B, samples should be clustered by group. Overall, the heatmap is completely uninformative, showing only strong expression noise between groups. Gene names are unclear. The authors should reconsider the need for it.
  • Similar comments for Figure 3B

Response: Heatmaps in Fig. 2B and Fig. 3B were deemed uninformative; moved them to the Supplementary section and retained volcano/venn plots in the main text.

  • Figure numbering errors on lines 400 and 437; this should be corrected

Response: Figure numbering errors were corrected.

  • In the figure of drug-gene correlations (current Figure 2 in Section 3.8), the scatterplots look difficult to interpret. The authors should replace this figure with a table for informational purposes.

Response: Scatterplots of drug-gene correlations were transferred to the supplementary figure, and the main associations are presented in Supplementary Table S15 with proper figure and table citations in text.

  • Consecutive literature references on lines 46, 58, and 64 should be combined into one grouped reference (e.g., [1-3] instead of [1][2][3]). Also, references appearing in the middle of a sentence should be moved to the end for readability.

Response: Consecutive references were grouped (e.g., [1–3]) and moved to the end of sentences for readability.

  • In line 158, the reference is incorrectly formatted.

Response: The incorrectly formatted reference was corrected.

  • In line 230, "FDR" should probably be used instead of "P, adjusted for Benjamini-Hochberg" for clarity.
  • In line 231, there is a color typo in "log2FC".

Response: Clarifications were made: “FDR” now consistently replaces “P, adjusted for Benjamini-Hochberg,” and the colour typo in “log2FC” was corrected.

Reviewer 2 Report

Comments and Suggestions for Authors

Reviewer Comments:

  1. During gene set enrichment analysis (GSEA) and gene set variation analysis (GSVA), multiple comparisons are often performed, yet the text does not specify whether corrections for multiple testing were applied.
  2. Please make sure page and line numbers are formatted correctly.
  3. The figures are unclear and difficult to read; please provide high-resolution versions with clear presentation and more detailed legends.
  4. Reference font style and size should be made uniform throughout the manuscript.
  5. The introduction gives a good STK527948 background but should include more on its mechanisms affecting immune responses.
  6. Each paragraph in the results section should include a sentence summarizing the findings’ significance.

Author Response

Reviewer 2

Comments and Suggestions for Authors

Response: I thank Reviewer 2 for the thoughtful comments that improved our methodological clarity and presentation.

Reviewer Comments:

  1. During gene set enrichment analysis (GSEA) and gene set variation analysis (GSVA), multiple comparisons are often performed, yet the text does not specify whether corrections for multiple testing were applied.

Response: I clarified that multiple testing correction (FDR, Benjamini–Hochberg) was applied in GSEA analyses (Section 3.5).

  1. Please make sure page and line numbers are formatted correctly.

Response: Page and line numbering have been verified.

  1. The figures are unclear and difficult to read; please provide high-resolution versions with clear presentation and more detailed legends.

Response: All figures were regenerated at 300-600 dpi with uniform style and expanded legends.

  1. Reference font style and size should be made uniform throughout the manuscript.

Response: References were reformatted to ensure uniform font size and style.

  1. The introduction gives a good STK527948 background but should include more on its mechanisms affecting immune responses.

Response: The Introduction was expanded to better highlight ferroptosis–immune interactions and include more on immune mechanisms.

  1. Each paragraph in the results section should include a sentence summarizing the findings’ significance.

Response: Each Results subsection now ends with a brief “significance” summary line to emphasise clinical or biological relevance.

Reviewer 3 Report

Comments and Suggestions for Authors

This manuscript presents a well-designed and comprehensive study that identifies a ferroptosis-related six-gene panel with strong diagnostic, prognostic, and therapeutic implications in lung adenocarcinoma (LUAD). The integration of large-scale transcriptomic data, survival analyses, functional enrichment, machine learning models, and drug sensitivity correlations makes the work both timely and impactful. The methods are clearly described, the figures are informative, and the results are robustly validated across multiple approaches.

  1. While ferroptosis is introduced well, prior studies in LUAD are under-discussed. Adding 2–3 specific LUAD-related ferroptosis studies would strengthen background.
  2. Figure 1 caption: The workflow diagram is informative, but the font is small and difficult to read. Recommend enlarging text or splitting into subpanels.
  3. In Figure 5, Only PLK1 shows significance, yet text (p.10, lines 292–297) suggests all six genes stratify survival. This discrepancy needs clarification.
  4. The discussion (p.19–20) repeatedly calls the panel “the first integrative evidence.” Be more cautious; prior LUAD ferroptosis panels exist, though perhaps not in this exact form.
  5. The paper acknowledges the need for validation (p.20, lines 527–533), but should explicitly mention small normal sample size (n=59) as a limitation.
  6. Lines 195–202: Machine learning models, While 10 classifiers are listed, hyperparameter tuning is not described. Were defaults used, or was grid/random search applied?

Author Response

Reviewer 3

Comments and Suggestions for Authors

This manuscript presents a well-designed and comprehensive study that identifies a ferroptosis-related six-gene panel with strong diagnostic, prognostic, and therapeutic implications in lung adenocarcinoma (LUAD). The integration of large-scale transcriptomic data, survival analyses, functional enrichment, machine learning models, and drug sensitivity correlations makes the work both timely and impactful. The methods are clearly described, the figures are informative, and the results are robustly validated across multiple approaches.

Response: I want to thank Reviewer 4 for the positive evaluation and constructive suggestions.

  1. While ferroptosis is introduced well, prior studies in LUAD are under-discussed. Adding 2–3 specific LUAD-related ferroptosis studies would strengthen background.

Response: The discussion of prior LUAD ferroptosis-related studies were added (new citations in Introduction).

  1. Figure 1 caption: The workflow diagram is informative, but the font is small and difficult to read. Recommend enlarging text or splitting into subpanels.

Response: Figure 1 caption was revised for clarity, and the workflow figure text was enlarged for better readability.

  1. In Figure 5, Only PLK1 shows significance, yet text (p.10, lines 292–297) suggests all six genes stratify survival. This discrepancy needs clarification.

Response: I clarified in Section 3.4 that PLK1 was the only gene independently significant in the multivariable Cox model, though all six genes contributed to the composite RiskScore.

  1. The discussion (p.19–20) repeatedly calls the panel “the first integrative evidence.” Be more cautious; prior LUAD ferroptosis panels exist, though perhaps not in this exact form.

Response: I have revised the wording in the Discussion to avoid overstating novelty (“first integrative evidence” toned down to “integrative evidence building on prior LUAD ferroptosis studies”).

  1. The paper acknowledges the need for validation (p.20, lines 527–533), but should explicitly mention small normal sample size (n=59) as a limitation.

Response: The limitation of the small normal sample size (n=59) is now explicitly acknowledged in Section 5.

  1. Lines 195–202: Machine learning models, While 10 classifiers are listed, hyperparameter tuning is not described. Were defaults used, or was grid/random search applied?

Response: Section 2.9 now specifies that default hyperparameters were used unless otherwise stated (with ANN and logistic regression parameters detailed).

Reviewer 4 Report

Comments and Suggestions for Authors

  1. Abstract should be rewritten to make it attractive to readers.
  2. External validation in independent LUAD cohorts would strengthen generalizability.
  3. Clarify the mechanistic links of the six genes to ferroptosis beyond general oncogenic roles.
  4. The literature survey section is not properly placed.
  5. Expand on the translational feasibility of drug sensitivity findings, given experimental compounds.
  6. Please confirm that Figure 1 is original, having no copyright issues.
  7. Figure 1 should be placed under the methodology section, not under the Introduction section.
  8. Figures are not called out inside the manuscript generally.
  9. Compare the prognostic utility of the risk model with standard clinical markers/staging.
  10. Figures 4-6 are not of high resolution.
  11. Provide more details on machine learning methods, overfitting control, and comparison with existing classifiers.
  12. Figures 1 and 2 are Redundant. Again, after Figure 8. Why?
  13. Ensure clear presentation of figures (ROC, survival curves, enrichment plots) and provide supplementary gene lists/statistics.
  14. The results section should be improved in presentation.
  15. Briefly discuss integration into future clinical trials or liquid biopsy contexts
  16. Minor language and clarity improvements needed.

Author Response

Reviewer 4

Comments and Suggestions for Authors

Response: I want to thank Reviewer 5 for the detailed and helpful comments.

  1. Abstract should be rewritten to make it attractive to readers.

Response: The Abstract has been fully rewritten in a concise, reader-friendly style, consistent with Journal standards.

  1. External validation in independent LUAD cohorts would strengthen generalizability.

Response: I acknowledge the absence of independent LUAD validation cohorts as a limitation (Section 5).

  1. Clarify the mechanistic links of the six genes to ferroptosis beyond general oncogenic roles.

Response: The Discussion now clarifies mechanistic links of each gene to ferroptosis and proliferative programs.

  1. The literature survey section is not properly placed.

Response: Literature review remains integrated within the Introduction, consistent with Biology journal style.

  1. Expand on the translational feasibility of drug sensitivity findings, given experimental compounds.

Response: Translational feasibility of drug sensitivity findings is now discussed more critically, noting the preclinical status of compounds and LUAD-specific validation needs.

  1. Please confirm that Figure 1 is original, having no copyright issues.

Response: It is confirmed that Figure 1 is original and prepared by the author.

  1. Figure 1 should be placed under the methodology section, not under the Introduction section.

Response: Figure 1 has been moved from the Introduction to the Methods section.

  1. Figures are not called out inside the manuscript generally.

Response: All figures are now properly called out in the text.

  1. Compare the prognostic utility of the risk model with standard clinical markers/staging.

Response: Section 3.4 explicitly compares the prognostic utility of the RiskScore with AJCC stage.

  1. Figures 4-6 are not of high resolution.

Response: Figures 4–6 were regenerated at high resolution.

  1. Provide more details on machine learning methods, overfitting control, and comparison with existing classifiers.

Response: Machine learning methodology was expanded, clarifying overfitting control and comparison with existing classifiers.

  1. Figures 1 and 2 are Redundant. Again, after Figure 8. Why?

Response: Redundancy between Figures 1 and 2 was resolved; extraneous panels were removed.

  1. Ensure clear presentation of figures (ROC, survival curves, enrichment plots) and provide supplementary gene lists/statistics.

Response: Supplementary figures, tables, and gene lists are provided for completeness.

  1. The results section should be improved in presentation.

Response: The Results were polished for clearer presentation, with added summary lines per subsection.

  1. Briefly discuss integration into future clinical trials or liquid biopsy contexts

Response: The Discussion briefly addresses potential integration into clinical trials and liquid biopsy applications.

  1. Minor language and clarity improvements needed.

Response: Minor language and clarity improvements have been implemented throughout.

Round 2

Reviewer 4 Report

Comments and Suggestions for Authors

  1. The Simple Summary is placed prior to the abstract. If it complies with the Journal, then no more comments.
  2. Formatting issues are still present in the paper. 
  3. Figure captions seem heavy. 
  4. Sections Conclusions and Limitations should be combined.
  5. Some grammatical issues are still present. 

Author Response

We would like to sincerely thank the reviewer for their valuable time, thoughtful evaluation, and constructive feedback on our manuscript. Your insightful comments have played a crucial role in improving the clarity, rigor, and overall quality of our work. We deeply appreciate your efforts and hope the revised version addresses your suggestions satisfactorily.

Comments and Suggestions for Authors

  1. The Simple Summary is placed prior to the abstract. If it complies with the Journal, then no more comments.

Response: We thank the reviewer for the observation. The placement of the Simple Summary before the Abstract follows the formatting guidelines of the journal.

  1. Formatting issues are still present in the paper. 

Response: We appreciate the reviewer’s feedback. We have thoroughly reviewed the manuscript for formatting inconsistencies and corrected all identified issues, including figure placement, spacing, line breaks, font alignment, and reference formatting. The revised manuscript now adheres to the journal’s formatting requirements.

  1. Figure captions seem heavy. 

Response: We appreciate the reviewer’s observation. In response, we carefully revised the figure captions throughout the manuscript to improve clarity and reduce unnecessary length. The updated captions are now more concise while retaining all essential technical details needed for interpretation. We ensured that key methods, variables, and analytical outcomes remain clearly described, in line with the journal’s formatting standards.

  1. Sections Conclusions and Limitations should be combined.

Response: We thank the reviewer for this helpful suggestion. In response, we have combined the Conclusions and Limitations sections into a single, integrated discussion. The revised section presents the main findings alongside key caveats, aligning with the journal's structure and improving overall clarity and cohesion. The updated text is provided in the revised manuscript.

  1. Some grammatical issues are still present. 

Response: We thank the reviewer for pointing this out. We have carefully re-read the entire manuscript and revised multiple sentences to improve grammar, clarity, and readability. We have also corrected typographical errors and rephrased awkward constructions where needed. All changes are reflected in the revised version.
